# Atomistic-Scale Energetic Heterogeneity on a Membrane Surface

**DOI:** 10.3390/membranes12100977

**Published:** 2022-10-07

**Authors:** Shiliang (Johnathan) Tan, Chisiang Ong, Jiawei Chew

**Affiliations:** 1School of Chemical and Biomedical Engineering, Nanyang Technological University, Singapore 637459, Singapore; 2Singapore Membrane Technology Centre, Nanyang Environmental and Water Research Institute, Nanyang Technological University, Singapore 637141, Singapore

**Keywords:** membrane filtration, interaction energy, energetic topology, molecular computation, nano-scale heterogeneity

## Abstract

Knowing the energetic topology of a surface is important, especially with regard to membrane fouling. In this study, molecular computations were carried out to determine the energetic topology of a polyvinylidene fluoride (PVDF) membrane with different surface wettability and three representative probe molecules (namely argon, carbon dioxide and water) of different sizes and natures. Among the probe molecules, water has the strongest interaction with the PVDF surface, followed by carbon dioxide and then argon. Argon, which only has van der Waals interactions with PVDF, is a good probing molecule to identify crevices and the molecular profile of a surface. Carbon dioxide, which is the largest probing molecule and does not have dipole moment, exhibits similar van der Waals and electrostatic interactions. As for water, the dominant attractive interactions are electrostatics with fluorine atoms of the intrinsically hydrophobic PVDF membrane, but the electrostatic interactions are much stronger for the hydroxyl and carboxyl groups on the hydrophilic PVDF due to strong dipole moment. PVDF only becomes hydrophilic when the interaction energy is approximately doubled when grafted with hydroxyl and carboxyl groups. The energetic heterogeneity and the effect of different probe molecules revealed here are expected to be valuable in guiding membrane modifications to mitigate fouling.

## 1. Introduction

The free energy of interfacial interactions is well-acknowledged to govern membrane fouling behaviors, with attraction and repulsion foulant–membrane interactions tied to more and less severe fouling, respectively [1,2]. On top of the fouling mitigation strategy of fabricating low-energy membranes, the energetic topology of the membrane surface has been further tied to the inherent surface roughness of polymeric membranes, which implies the importance of local rather than overall interactions [3,4]. Specifically, such physical heterogeneity of a membrane surface plays a significant role due to its influence on the settlement behavior of the foulants onto the membrane [5], as well as the interaction energy between the foulants and the membrane surface [6]. Understanding the energetic topology would also aid in the modification of membranes to mitigate fouling.

Atomic force microscopy (AFM) is commonly experimentally employed to quantify the surface roughness of membranes and also the interaction forces between the probing cantilever tip and the surface being characterized. The applied forces between the tip and surface can be used for the detection of mechanical magnetic, van der Waals, electrostatic, capillary, chemical and steric contact forces [7], providing a powerful tool for the characterization of membranes and thereby predicting membrane fouling tendencies. Furthermore, the classical Derjaguin–Landau–Verwey–Overbeek (DLVO) and extended DLVO (XDLVO) theories are commonly used to quantify the surface free energy based on non-covalent interactions, namely Lifshitz–van der Waals, Lewis acid–base, electrostatic double layer and Brownian forces [8,9]. Foulant–membrane and foulant–foulant interfacial forces have been measured experimentally and proven to correlate well with membrane fouling behaviors [10,11,12]. Another technique, namely surface element integration (SEI), has also been developed to determine the interaction energy between a particle and membrane surface morphology [13]. Hoek and Agarawal applied SEI by integrating it with the AFM technique to produce a distribution of interaction energy profiles of polyamide membranes with alumina, silica and latex particles [4]. However, while the above-mentioned techniques allow for interaction energies to be revealed, the molecular-level mechanisms underlying such energetics remains unknown.

To this end, energetic sampling is a promising technique to evaluate the roles of molecules in determining the energetic topology of the membrane [14]. Such interaction energies, which involve the van der Waals and electrostatic constituents, are directly governed by the force field parameter characteristics of the molecules investigated [15,16,17]. Therefore, the objective of this study is to propose a simple computational method to determine the energetic topology of a common polymeric membrane. This is illustrated with polyvinylidene fluoride membrane (PVDF), both hydrophobic and hydrophilic, and three representative probe molecules (namely argon, carbon dioxide and water) at the atomistic level, which is not possible experimentally.

## 2. Method

A 10 nm by 10 nm by 1 nm polyvinylidene fluoride (PVDF) membrane with a density of 1.75 g/cm^3^, which is the same as that reported in Velioglu et al. [18], was used in this study, with the force field parameters adopted from Lachet et al. [19]. Per that reported earlier [20], the PVDF was made hydrophilic by adding 98 hydroxyl and 98 carboxyl functional groups randomly onto the PVDF surface to mimic the hydrophilic PVDF used experimentally. Three probe molecules were evaluated, namely argon, carbon dioxide and water, which represent different sizes and natures. Argon atoms are spherical and thus only interact via van der Waals [21]. As for carbon dioxide, it is a linear molecule, and thus both van der Waals and electrostatic interactions are equally dominant. Regarding water, interactions are predominantly electrostatic [22]. The force field parameter of argon was modeled by Vrebec et al. [23], carbon dioxide by Potoff and Siepmann [24] and water by TIP4P (rigid planar four-site interaction potential) [25], which are presented in Table 1.

The intermolecular interaction energy, *u*, for the *i*th molecule is calculated by summing all site interactions of the molecule with that on PVDF, which are contributed by the van der Waals and electrostatic interactions:$$u_{i,PVDF}=u_{i,PVDF}^{electrostatic}+u_{i,PVDF}^{VdW}$$

The van der Waals were described by 12-6 Lennard-Jones:(1)$$u_{i,PVDF}^{VdW}=4\sum_{\alpha =1}^{A}\sum_{\beta =1}^{Β}\varepsilon_{i,PVDF}^{\alpha ,\beta }\left[{\left(\frac{\sigma_{i,PVDF}^{\alpha ,\beta }}{r_{i,PVDF}^{\alpha ,\beta }}\right)}^{12}-{\left(\frac{\sigma_{i,PVDF}^{\alpha ,\beta }}{r_{i,PVDF}^{\alpha ,\beta }}\right)}^{6}\right]$$
where *A* is the number of sites on molecule *i*, and *B* is the number of sites on PVDF. The parameter $X_{i,j}^{\alpha ,\beta }$ is associated with site *α* in molecule *i* and site *β* in PVDF. The parameter $r_{i,PVDF}^{\alpha ,\beta }$ is the distance between the two sites, and $\sigma_{i,PVDF}^{\alpha ,\beta }$ and $\varepsilon_{i,PVDF}^{\alpha ,\beta }$ are the cross-collision diameter and the cross-well depth of interaction energy, respectively, and are calculated from the Lorentz–Berthelot rule:(2)$$\sigma_{i,PVDF}^{\alpha ,\beta }=\left(\sigma_{i}^{\alpha }+\sigma_{PVDF}^{\beta }\right)/2\;\;\;\;\;\;\varepsilon_{i,PVDF}^{\alpha ,\beta }=\sqrt{\left(\varepsilon_{i}^{\alpha }\;\varepsilon_{PVDF}^{\beta }\right)}$$

The electrostatic interactions are described by the Coulombic potentials:(3)$$u_{i,PVDF}^{electrostatic}=\frac{e^{2}}{4\pi \varepsilon_{0}}\sum_{a=1}^{A}\sum_{b=1}^{B}\frac{q_{i}^{a}q_{PVDF}^{b}}{r_{i,PVDF}^{a,b}}$$
where *q_i_* are partial charges assigned to sites on the molecules.

The energetic topology of the membrane was obtained by freezing the atoms of the PVDF membrane in place while the probing molecules were added into a lattice with 0.05 nm spacing in the x and y direction, and 0.02 nm in the z direction, as illustrated in Figure 1. In particular, for carbon dioxide and water, whereby the orientations have been reported to have significant impacts on interaction energy [27], the molecules were given 1000 random orientations with respect to their mass centers. At a given lattice point, the minimum energy and the z-distance were recorded for the construction of the energetic profile, as illustrated in Figure 1c. This would provide the interaction energy (constituted by van der Waals and electrostatic interactions) profile between each probe molecule and the PVDF membrane.

## 3. Results and Discussion

Figure 2a–c shows the distance profile at which the minimum interaction energy between the probe molecule and the surface are located. The distance profile shows a similar trend between different probing molecules, where the profiles show the physical representation of the PVDF atoms, similar to what is shown in Figure 1c. The minimum energy profiles at these distances are presented in Figure 2d–f. Two observations are noteworthy. Firstly, irrespective of the probe molecule, both the distance and energy profiles are clearly heterogeneous. This indicates nano-scale variations in the energetics, which is on top of the micro-scale ones revealed via AFM earlier [28] and the nano-scale surface roughness also revealed via AFM [4]. Secondly, the distance profiles and energy profiles are not directly related. Although distance profiles are similar to one another, the energy profiles vary significantly, implying the probe molecule’s nature significantly affects the profiles. Not only are the energy magnitudes different, but the locations of energy wells (i.e., most negative energies) are also different between charged and uncharged molecules. Specifically, water has the strongest interaction, followed by carbon dioxide and argon.

Although the size of the probe molecule affects the ability to probe into small crevices, these profiles are dependent on the interaction between the probe and membrane. This can be observed from Figure 2a–c, where the blue area (closest to the surface) located between PVDF atoms is largest for stronger interacting probes and not the size of the probe molecule, which follows the order of water, carbon dioxide and argon. In contrast, the probe molecules at the crevice also have the lowest interaction, as they can interact with the most atoms, as shown in Figure 2d–f, indicating that the interacting energy is directly proportional to the density of PVDF membrane atoms. In addition, these interactions depend on the probing molecule, as these figures show that strong interaction occurs on the region with most fluorine atoms for charged molecules (carbon dioxide and water), whereas strong interacting regions occur in the crevice for the case of argon.

Note that argon does not have electrostatic interaction energy due to the atom being of neutral charge. In addition, due to the lack of electrostatic interaction and being a sole atom, the probe molecule argon was able to produce a similar distance and energy profile. These profiles describe the atomistic heterogeneity of the membrane, and the 3D image is illustrated in Figure 2g.

To further understand the energy profiles of carbon dioxide and water, the profiles of the van der Waals and electrostatic interaction energy constituents are presented in Figure 3. Figure 3a,c show that the van der Waals interaction magnitude is the greatest for carbon dioxide, followed by water, due to the increased number of atoms in the probe molecule. In contrast, the electrostatic interaction profiles in Figure 3b,d show the opposite, where the interaction energy is greater for water than carbon dioxide due to the strong dipole moment of water, whereas the dipole moment is neutralized by its dipole being a linear molecule. Moreover, Figure 3b,d also indicate that fluorine atoms are responsible for the high electrostatic interactions, and these interactions increase with strong dipole moment, as for the case of water.

Although water has relatively high interaction with fluorine atoms, PVDF is still classified as a hydrophobic membrane. This shows that the interaction is still unable to overcome the energy barrier for water to wet the surface as most areas of the membrane are similar or have lower interaction energy than a water molecule in the liquid phase. In particular, the interaction of a pair of water molecules is −25 kJ/mol; as there are about four neighboring water molecules, it would require about −100 kJ/mol for the membrane to be hydrophilic [26]. Here, we show the distance and minimum energy profile of hydrophilic PVDF in Figure 4. It is to no surprise that the distance profile in Figure 4a represents the physical location of the atoms of the PVDF, as was shown in previous cases. The energy profile in Figure 4b shows a much stronger interaction with the hydroxyl and carboxyl group as compared to the fluorine atoms, where the energy in those regions is almost doubled. This strong interaction is the main reason that this PVDF membrane is hydrophilic. Although it is known that hydroxyl and carboxyl groups have a high affinity with water, this case is used to illustrate the application of the probing method to understand the atomistic interaction with a probe molecule. It would be useful to probe a lesser-known material and to identify the stronger interaction region for the purpose of surface modification.

## 4. Conclusions

The molecular computation results demonstrate that the atomistic-scale surface energetics are clearly heterogeneous, and the magnitudes and gradients depend significantly on the probe molecules. The interaction energy profiles provide molecular insights as to why various regions of the membrane have relatively stronger or weaker interactions. The most significant total interaction is exhibited by water, which is tied to strong electrostatics interaction, while the weakest is exhibited by argon. Considering argon, which only has van der Waals interactions, attractive interactions are formed with all the PVDF molecules on the surface, and thus the regions with denser PVDF molecules exhibit higher energy magnitudes. Probe molecules with charge, i.e., carbon dioxide and water, are strongly attracted to regions with high local densities of fluorine atoms, resulting in higher interaction energies. Carbon dioxide, the largest probing molecule, interacts with PVDF through similar contributions from both van der Waals and electrostatic interactions. Regarding water, which has the strongest electrostatic interactions, the interaction is significant with the intrinsically hydrophobic PVDF membrane, and the interaction energy is further doubled in the presence of hydroxyl and carboxyl for the case of the hydrophilic PVDF. This method allows for the understanding of the energetic topology of a surface such as a membrane, and it is expected to be valuable in the surface modification of new materials.

## Figures and Tables

**Figure 1 membranes-12-00977-f001:**
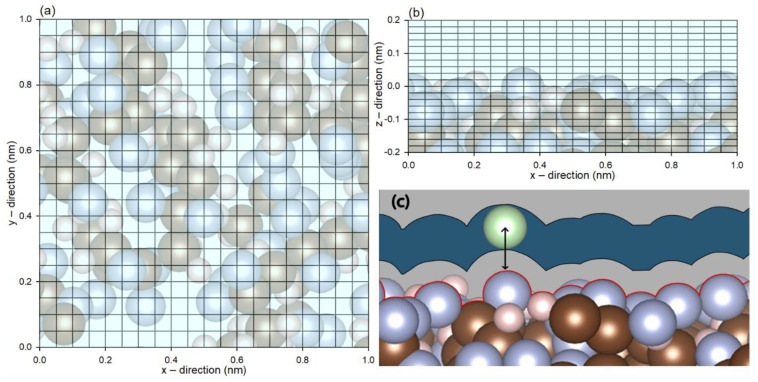
Coordinates of the probing molecules in the lattice spacing from the (**a**) top and (**b**) side views. (**c**) Schematic diagram of the distance between a probing molecule and the membrane.

**Figure 2 membranes-12-00977-f002:**
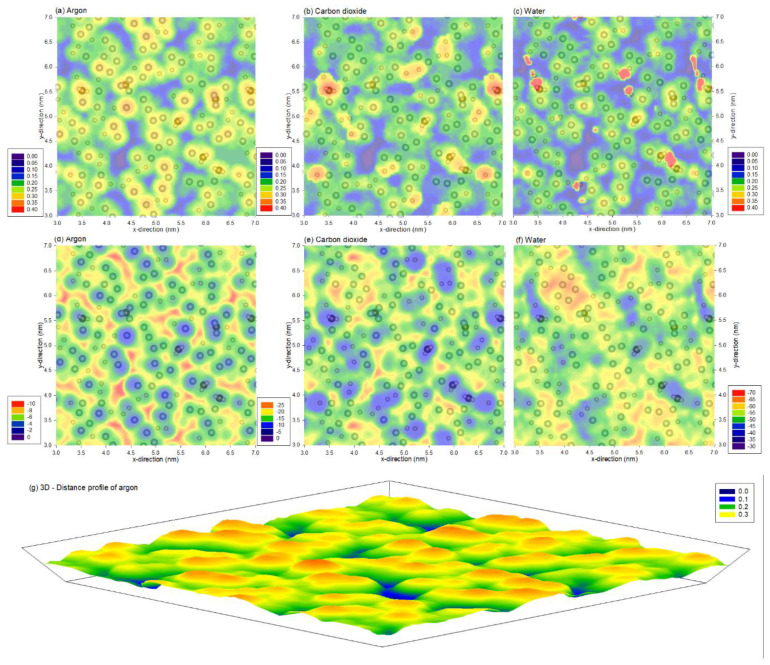
(**a**,**b**) distance of the probe molecule at the (**d**–**f**) minimum total interaction energy between (**a**,**d**) argon (**b**,**e**) carbon dioxide and (**c**,**f**) water molecules and the PVDF membrane. Note that the profiles are superimposed with PVDF atoms from the top 0.1 nm of the membrane, and the range for the legends are different, represented in (**a**–**c**) nm and (**d**–**f**) kJ/mol. (**g**) represents (**a**) in three-dimensional form.

**Figure 3 membranes-12-00977-f003:**
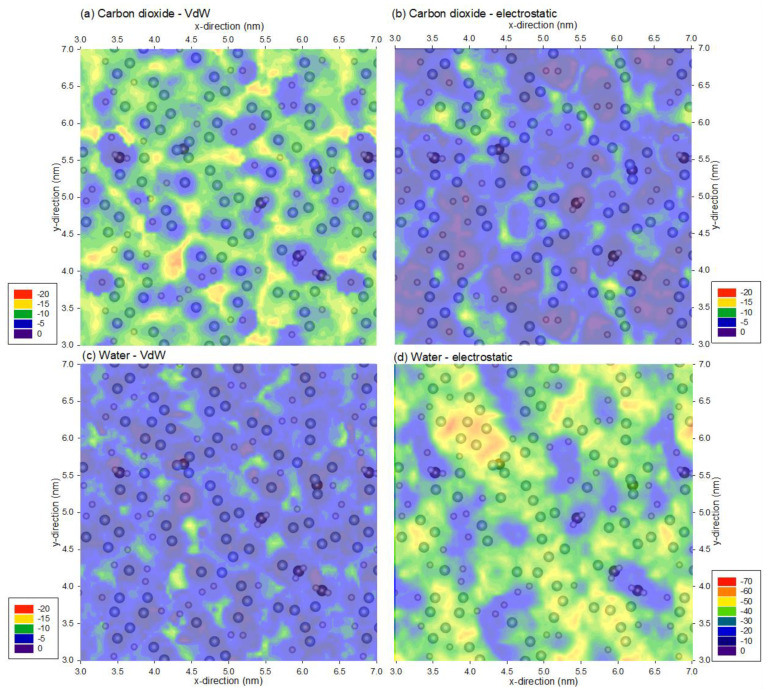
Minimum van der Waals and electrostatic interaction energy of (**a**,**b**) a carbon dioxide and (**c**,**d**) water molecule and the PVDF membrane. Note that the profiles are superimposed with PVDF atoms from the top 0.1 nm of the membrane, and the range for the legends are different; the legends are represented in kJ/mol.

**Figure 4 membranes-12-00977-f004:**
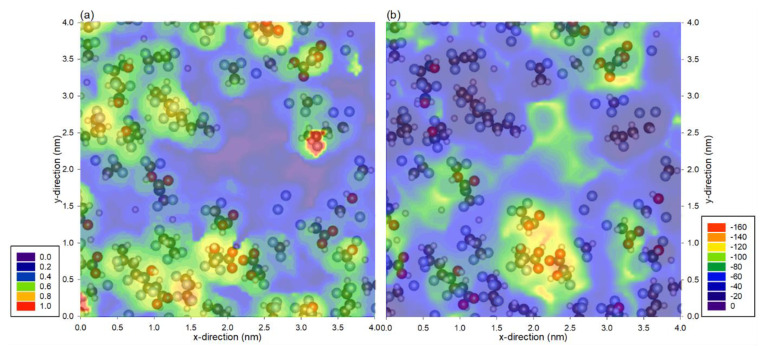
(**a**) Distance of the probe molecule and the (**b**) minimum total interaction energy between water probe molecule and the hydrophilic PVDF membrane. Note that the profiles are superimposed with hydrophilic PVDF atoms from the top 0.2 nm of the membrane, and the legends are represented in (**a**) nm and (**b**) kJ/mol.

**Table 1 membranes-12-00977-t001:** Force field parameters of the probe molecules.

Fluid	Parameter	Symbols	Unit	Value
Argon [23]	Ar	*σ*	nm	0.33952
		*ε*/*k*	K	116.79
Carbon dioxide [24]	C	*σ*	nm	0.28
		*ε*/*k*	K	27
		*q*	e	0.7
	O*	*σ*	nm	0.305
		*ε*/*k*	K	79
		*q*	e	−0.35
	Atomic distance	C=O	nm	0.116
Water, TIP4P [26]	O	*σ*	nm	0.311831
		*ε*/*k*	K	208.08
		q	e	−0.8391
	H	q	e	0.41955
	Atomic distance	O-H	nm	0.11549
	Angle	H-O-H	o	104.52

## Data Availability

Not applicable.

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
