# Peer review of "Atomistic-Scale Energetic Heterogeneity on a Membrane Surface"

_membranes, 2022, doi:10.3390/membranes12100977_

Round 1

Reviewer 1 Report

In the current report, Chew and coworkers have presented the results of Atomistic-scale energetic heterogeneity on a membrane surface based on their developed model. I think this method can be extensively applied to the anti-fouling membrane design where: 1. the chemistry of the membrane can be easily tuned; 2. The changes in charge fraction or functional group fraction; and 3. The types of the probe molecules. I am generally positive for the publication of this work after addressing the following concerns:

1.     The methodology part is not clear enough. If I understand the method correctly, I think the presented method is similar to the Monte Carlo method. For a given probe molecule type, is that true for each of the grid lattice on the X-Y plane, 1000 trials have been applied with changing the orientation of the probe molecule? If it is true, how do the authors deal with the depth in the Z-direction? Meaning at what specific distance the “insertion” has been performed? Thus, the method part has to be improved as explicitly as possible.

2.     From line 147 to 149: “Figure 3a and c show that the van der Waals interaction magnitude is the greatest for carbon dioxide, followed by water, due to the increased number of atoms in the probe molecule.” Is this a valid statement? The carbon dioxide has the same number of atoms as water molecule. If the authors are referring to the number of the particles in the TIP4P model, the statement cannot be hold as well since only the Oxygen atom has the VDW interaction with other atoms. Could the authors clarify on this point?

3.     Following by the above point, it is inconsistent for the description of the water model in Line 77 (SPC/E) and in Table 1 (TIP4P).

4.     Maybe I have missed it, but the composition differences between the hydrophobic and hydrophilic PVDF are not introduced, what is the number of the hydroxyl and carboxyl groups that were added on to the PVDF?

Minors:

1.     In the caption of Figure 2, subplot (g) has not been introduced.

2.     Line 173: “this case is used to illustrate the application of the probing method to understanding …” should be “this case is used to illustrate the application of the probing method to understand …”

3.     Line 178: “Figure 4. (a) distance of the probe molecule at the (b) minimum” should be “Figure 4. (a) distance of the probe molecule and the (b) minimum”;

4.     Line 181: “represented in (a) nm and (b) kJ/mol..” should be “represented in (a) nm and (b) kJ/mol.”

5.     Line 199: “and is expected to be valuable in the surface modification of new materials..” should be “and is expected to be valuable in the surface modification of new materials.”

Author Response

In the current report, Chew and coworkers have presented the results of Atomistic-scale energetic heterogeneity on a membrane surface based on their developed model. I think this method can be extensively applied to the anti-fouling membrane design where: 1. the chemistry of the membrane can be easily tuned; 2. The changes in charge fraction or functional group fraction; and 3. The types of the probe molecules. I am generally positive for the publication of this work after addressing the following concerns:

 Response:

We thank the reviewer for the detailed comments, which have helped to improve the manuscript significantly.

Comment 1

The methodology part is not clear enough. If I understand the method correctly, I think the presented method is similar to the Monte Carlo method. For a given probe molecule type, is that true for each of the grid lattice on the X-Y plane, 1000 trials have been applied with changing the orientation of the probe molecule? If it is true, how do the authors deal with the depth in the Z-direction? Meaning at what specific distance the “insertion” has been performed? Thus, the method part has to be improved as explicitly as possible.

Response:

The sampling space is divided into grid lattices in the X-, Y-, and Z-direction of 0.05 nm, 0.05 nm, and 0.02 nm, respectively. This is specified on Page 5:

“…0.05 nm spacing in the x and y direction, and 0.02 nm in the z direction…”

For every grid lattice on the Z-direction, the probe molecule is sampled on the grid lattice in X-Y plane, and were given 1000 random orientations. This is specified on Page 5:

“…molecules were given 1000 random orientations with respect to their mass centers…”

Comment 2

From line 147 to 149: “Figure 3a and c show that the van der Waals interaction magnitude is the greatest for carbon dioxide, followed by water, due to the increased number of atoms in the probe molecule.” Is this a valid statement? The carbon dioxide has the same number of atoms as water molecule. If the authors are referring to the number of the particles in the TIP4P model, the statement cannot be hold as well since only the Oxygen atom has the VDW interaction with other atoms. Could the authors clarify on this point?

Response:

We affirm that the Van der Waals (VdW) interaction does increase with the increase in the number of atoms in the probe molecule.

This also depends on the size of the atom as well, as a larger atom would exert a higher interaction. In the case of water, only the Oxygen atom of water exerts VdW interaction because the Hydrogen atom is very small, and thus would exert a very low VdW interaction. Additionally, it is well within the collision diameter of the Oxygen atom, therefore, the VdW contribution from the Hydrogen atom is negligible. This is the same for every type of water model. Even though the TIP4P model has four charge points or four particles, these charge points do not contribute to the VdW interaction.

Comment 3

Following by the above point, it is inconsistent for the description of the water model in Line 77 (SPC/E) and in Table 1 (TIP4P).

Response:

TIP4P water model was used for the computation. This has been added on Page 4:

“…TIP4P (rigid planar four-site interaction potential)…”

Comment 4

Maybe I have missed it, but the composition differences between the hydrophobic and hydrophilic PVDF are not introduced, what is the number of the hydroxyl and carboxyl groups that were added on to the PVDF?

Response:

We apologize for the omission. The following has been added on Page 3: “…98 hydroxyl and 98 carboxyl functional groups randomly onto the PVDF surface to mimic the hydrophilic PVDF used experimentally…”. This is determined from the XPC spectrum, which shows the amount of oxygen concentration.

Minor Comments

  1. In the caption of Figure 2, subplot (g) has not been introduced.
  2. Line 173: “this case is used to illustrate the application of the probing method to understanding …” should be “this case is used to illustrate the application of the probing method to understand …”
  3. Line 178: “Figure 4. (a) distance of the probe molecule at the (b) minimum” should be “Figure 4. (a) distance of the probe molecule and the (b) minimum”;
  4. Line 181: “represented in (a) nm and (b) kJ/mol..” should be “represented in (a) nm and (b) kJ/mol.”
  5. Line 199: “and is expected to be valuable in the surface modification of new materials..” should be “and is expected to be valuable in the surface modification of new materials.”

Response:

We thank the reviewer for pointing these out, and have corrected all accordingly.

Reviewer 2 Report

The present work provides insight into the surface heterogeneity of hydrophobic and hydrophilic PVDF membranes. The interaction profiles vary for the probe molecules involved according on whether they are Ar, CO2 or H2O depending on the surface heterogeneity considered. These results are interesting and well supported. The methodology employed has been adequate. Therefore, publication in the present form is recommended.

Author Response

We thank the reviewer for the positive opinion of this manuscript.

Round 2

Reviewer 1 Report

The authors have addressed all my concerns, the paper could be accepted and published on Membranes.